# Robotic Surgery: Rediscovering Human Anatomy

**DOI:** 10.3390/ijerph182312744

**Published:** 2021-12-03

**Authors:** Antonio Gangemi, Betty Chang, Paolo Bernante, Gilberto Poggioli

**Affiliations:** 1Department of Surgery, University of Illinois at Chicago College of Medicine, Chicago, IL 60612, USA; 2University of Illinois at Chicago College of Medicine, Chicago, IL 60612, USA; bchang22@uic.edu; 3Center for Metabolic and Obesity Surgery, IRCCS Azienda Ospedaliera Universitaria di Bologna, 40121 Bologna, Italy; bernantepaolo@gmail.com; 4Department of Medical and Surgical Sciences, St.Orsola-Malpighi University Hospital of Bologna, 40138 Bologna, Italy; gilberto.poggioli@unibo.it

**Keywords:** robotic surgery, surgical anatomy, surgical anatomist

## Abstract

Since its advent, robotic surgery has redefined the operating room experience. It directly addressed and resolved many of the shortcomings of laparoscopic methods while maintaining a minimally invasive approach that brought benefits in cosmesis and healing for patients but also benefits in ergonomics and precision for surgeons. This new platform has brought with it changes in surgical training and education, principally through the utilization of virtual reality. Accurate depictions of human anatomy seen through augmented reality allow the surgeon-in-training to learn, practice and perfect their skills before they operate on their first patient. However, the anatomical knowledge required for minimally invasive surgery (MIS) is distinct from current methods of dissection and prosection that inherently cater towards open surgery with large cuts and unobstructed field. It is integral that robotic surgeons are also equipped with accurate anatomical information, heralding a new era in which anatomists can work alongside those developing virtual reality technology to create anatomical training curricula for MIS. As the field of surgery and medicine in general moves to include more and more technology, it is only fitting that the building blocks of medical education follow suit and rediscover human anatomy in a modern context.

## 1. The Advent of Robotic Surgery: A Mere Technological Innovation or Something More?

The introduction of laparoscopy in the 1980’s marked a major advancement in the surgical field. This minimally invasive approach permitted less post-operative pain, better cosmetic results, shorter hospital stays, and faster operation times than traditional open surgery [1,2]. These advantages led to the rapid and wide adoption of laparoscopy across the globe.

In the matter of a decade from its introduction, laparoscopy became the gold standard for a wide range of surgical procedures such as cholecystectomy, appendectomy, inguinal hernia repair, gastric fundoplication, adrenalectomy, bariatric surgery, colectomy, splenectomy, and nephrectomy [3].

As with any groundbreaking technology, the initial enthusiasm was followed by a more critical appraisal as its limitations became more readily apparent. Laparoscopic instruments are long and do not have tips that rotate with the same dexterity as the human wrist. Their manipulation is constrained by the “fulcrum effect” as the tip of the instrument moves in the opposite direction of the surgeon’s hand that is holding the laparoscopic instrument (Figure 1) [4].

Furthermore, the laparoscopic camera projects a 2D video feed on the laparoscopic monitor, requiring the surgeon’s brain to constantly adapt their 3D knowledge about anatomy and the surgical field into a 2D format (the laparoscopic screen) to orient themselves to their positioning in the body. In short, the laparoscopic surgeon is asked to master non-instinctual skills using tools that do not adequately mimic human dexterity or cognitive processing. In one word, laparoscopy is not “intuitive”. Due to these limitations, the learning curve of laparoscopy turned out to be long and steep [5,6]. Many surgeons felt discouraged while others adopted this approach to perform complex surgical procedures and were forced into uncomfortable postures, placing unnecessary mechanical strain on their wrists, arms, and shoulders to compensate for the engineering limitations of this technology. Almost every aspect of the operation—table height, monitor position, and instrument grip—opens the surgeon up to the risk of musculoskeletal strain [7]. Park et al. find that 86.9% of laparoscopic surgeons experience pain and discomfort attributable to their use of the laparoscopic approach [8]. It is based on these critical limitations that Drs. Moll, Freund and Robert created Intuitive Surgical (Sunnyvale, CA, USA) in the late 1990’s. This company attempted to combine tele-robotic technology and human–machine interfaces with minimally invasive surgery. In 2000, their groundbreaking and innovative robotic system, named fittingly after Leonardo da Vinci, received FDA approval for surgical use in the United States.

In the present day, the Da Vinci operating system is on its fourth generation [9]. The robot has eliminated virtually all limitations of laparoscopy by including a robotic camera that is capable of three-dimensional and fluorescence-aided vision, helping the surgeon differentiate and identify organs/landmarks with fluorescent markers (Figure 2) [10], robotic instruments that have seven degrees of freedom and lack any tremor (inescapable with any hand holding a surgical instrument) and the ability to scale movements of the human hand holding the master grips at the robotic console to a much finer and smaller size.

These striking technical advantages led to swift adoption of the robotic technology. This approach decreased the risk of open conversion both during complex, technically challenging surgical procedures and in severe conditions involving high risk patients [11,12,13]. With the expiration of Intuitive’s original patent, there has been a burgeoning release of newer robotic surgery platforms and companies, like Medtronic’s Hugo^®^ system, which are making robotic-assisted surgery more accessible than ever.

The robotic platform introduced crisp, high-quality three-dimensional vision with up to 10-fold magnification that enabled the surgeon to see details of the human anatomy that would otherwise have been inaccessible in the open setting without surgical loupes or a surgical microscope. In fact, this kind of detail would be near impossible to appreciate without a visual aid even in a prosection theater with a skilled and knowledgeable surgical anatomist. The computer-based nature of this technology also opens the door to novel applications of the study and teaching of human anatomy and is uncovering a new role for the surgical anatomist. Robotic surgery is enabling surgeons and anatomists to redefine the study of human anatomy.

## 2. Robotic Surgery and the Recreation of Surgical Anatomy in a Virtual Reality (VR) Environment

Robotic surgery has threatened the existing surgical dogma: the surgeon can now perform a surgical procedure without any physical contact with the patient. In fact, early prototypes for robotic surgery systems were created by the Department of Defense for trauma surgery at a distance and telesurgery [14]. The manipulation of the master grips at the working unit (the robotic “console”) by the operating surgeon generates high-frequency electrical signals that travel through co-axial cables to the robotic tower where are they are analyzed and filtered by the computer (“vision cart”) and then transferred through another set of co-axial cables to the remote unit (“patient side cart”) where the data are converted into movements of the robotic arms and the robotic instruments (Figure 3) [15].

Similarly, the information captured by the robotic camera and the robotic arms at the operating room table is being transferred to the robotic tower and then to the robotic console where the surgeon receives and acts on it. Altogether, these three components form a system that is constantly exchanging information bidirectionally through coaxial cables.

The same machine that affected the lives of virtually every human living in the 20th century—the computer—is now affecting modern day surgery as the training process to master this disruptive technology moves from operation and dissection to VR. The computer is the common denominator and link between the robotic system and its training surrogate: the robotic simulator. The trainee sits at a replica of the surgical console and is able to view and manipulate a virtual field. The computer of the robotic simulator recreates the surgical anatomy in VR while maintaining the realism of the robotic console’s hardware. Initially, various standalone simulators were introduced and scientifically validated for their face and content validity (Figure 4) [16].

More recently, the robotic industry has introduced a novel generation of robotic simulators that can be integrated with the robotic console, allowing trainees to safely hone their robotic surgery skills while sitting at the same console that they will be using when operating on human subjects (Figure 4a) [16].

The use of VR simulation in surgical training has already been shown to improve knowledge, self-confidence, and surgical skill [17,18,19]. In addition, there is already preliminary evidence indicating promising clinical applications of VR training for procedures such as cholecystectomy and knee arthroscopy [20,21,22]. The ability for a good training performance to translate to more successful clinical outcomes is what makes education through simulation a viable educational tool. The realism of the surgical anatomy being recreated with the computer software of robotic simulators is of critical importance because it allows learners to familiarize themselves with the human body and all of its intricacies from the unique vantage point of the robotic camera. A precise, truthful, and accurate pairing of the VR anatomy with real life anatomy and its most common variations is integral to the success of this technology. Surgical planning using VR technology has been shown in multiple disciplines to improve a surgeon’s spatial understanding of their surgical field and anatomical orientation [23,24]. Even VR simulation used in the context of medical school has been shown to augment knowledge of anatomy and visuospatial orientation [25]. The promising application of VR in surgical education suggests a preeminent role for the surgical anatomist, in collaboration with surgeons and software engineers, to improve anatomic fidelity in these learning tools. Through this partnership, the robotic simulator’s capacity to train on specific surgical procedures and assess the trainee’s competence and decision-making skill will continue to advance.

## 3. Robotic Surgery and the Traditional Teaching of Prosection and Dissection

Historians would generally agree that the birth of modern human anatomy can be traced back to the 16th century when Andreas Vesalius (University of Padua 1537–1542) authored an extremely influential book on human anatomy: De Humani Corporis Fabrica Libri Septem—*On the Fabric of the Human Body*. This masterpiece caused a shift in the medical doctrine from dogmatic teaching based on the reading of ancient books to observational learning based on the prosection and dissection of cadavers. Since then, the knowledge of human anatomy has progressed and novel techniques of preservation have been introduced, but prosection and dissection are still carried out by anatomists and students through an open approach where there is wide exposure of the various organs and structures, which trains students to operate using the same modality—open surgery. Robotic surgery, thanks to its numerous technical advantages, has broadened the applications of traditional minimally invasive surgery (laparoscopy) and has decreased the need to convert to an open approach.

The perspective of a student learning anatomy through open dissection of a donor differs vastly from the perspective provided to the minimally invasive surgeon. Dissection provides the learner with a large field of vision and information about the placement of certain structures relative to others. It lends a birds-eye-view and limits the dissector’s acquaintance of the human body to this plane of vision. When these students begin their training in robotic surgery, their entire perspective changes. Now, localization of the target organ requires navigation through the body cavity in a completely different plane. The camera limits the width of visual field and learners must use landmarks not previously noted during their dissection to arrive at their destination.

Additionally, both robotic and laparoscopic surgery require the insufflation of carbon dioxide or other gas inside the abdominal (pneumoperitoneum), thoracic, or other cavity of the human body to successfully expose the surgical field where the surgeon will operate. However, the insufflated gas increments of the pressure inside the cavity (commonly 12 to 15 mmHg for an adult patient) which will transfer to the wall and tissues of the various organs of the surgical field, producing a number of morphologic changes. For instance, the liver volume decreases and alterations in the diameter of vascular structures and intestines or displacement of the muscle diaphragm have been documented [26,27].

To our knowledge, prospective anatomists are not currently receiving any formal training to apply their anatomical knowledge to the setting of minimally invasive surgery. This is an area in which there are great strides to be made. The nuanced differences in these approaches to human anatomy and the need to adapt to them suggest a shift in culture among surgical anatomists who are called to reproduce the visuo-spatial conditions of the anatomy seen through the “artificial eye” of the robotic camera in the dissection and prosection theater.

## 4. Discussion: A Proposed Look Ahead

The role of anatomy in the education of future robotic surgeons is in two distinct stages: when students encounter anatomy in medical school and when surgeons train using the robotic simulator.

Students are first exposed to anatomical structures in the cadaver labs of medical schools in which open dissection is the prevailing method of education. As stated before, this is incongruous with the robotic approach and is an area in which surgical anatomists can better prepare students for a field shifting rapidly to less invasive methods. The anatomical changes seen with the use of gas in the body to increase visibility and space during minimally invasive procedures and the unique perspective of the robotic camera are significant and are not currently emphasized during the training of surgical anatomists. If those training to teach human anatomy are provided with this information and are able to see how structures are affected by the robotic approach, they would be better equipped to incorporate this information into the standard anatomical curriculum.

The next important use for relevant anatomical knowledge comes when aspiring robotic surgeons begin to train using VR technology. Current methods used in the development of VR-based training systems for surgeons involve a technique called “photogrammetry”. This method essentially allows the user to compile photos and videos taken from different angles and perspectives and uses common points between the images to create a 3D reconstruction of the field [28]. This 2D to 3D conversion creates a hyper-realistic reconstruction that stays true to the color, texture, and contours of the original structure. When applying this technique to anatomic representations in VR, different companies compile footage both in the form of videos and photos during different surgical operations and create 3D depictions of the anatomy in order to provide users with training material that is true to real human anatomy. The following is a possible role that surgical anatomist can play in this process.

In Petriceks et al. (2018), a limitation indicated was the effect of lighting and obstructions in the ability to correctly render and visualize certain reconstructed figures [28]. These are common situations in which guidance from an expert in anatomy could be extremely useful. In this circumstance, the role of the surgical anatomist would be to identify problem areas in the reconstruction and use their knowledge base, along with the skills of those building the 3D structures, to edit and add any missing details that would create an accurate rendering despite technological limitations.

In addition, surgical anatomists would be integral in the process of selecting both prototypical cases and cases in which there is some anatomical variation to present students with variation representative of their future patients. As efficient as photogrammetry is, it still requires the editorial eye of an anatomist to ensure that teaching material is accurate.

In many ways, this role necessitates the expertise of the anatomist. The robotic surgeon is well-versed in the skills and regional anatomy of the areas they are operating on, but this anatomical knowledge is based on the specific patient population, demographics, or other personal differences in their practice and can be unrepresentative of the breadth of possible anatomical differences. They may be less suited to develop a training tool meant to be broad in its user base of future robotic surgeons. It is in these nuances that surgical anatomists can lend their knowledge.

## 5. Conclusions

Since its advent, robotic surgery has redefined the operating room experience. It directly addressed and resolved many of the shortcomings of laparoscopic methods while maintaining a minimally invasive approach that saw improvements in cosmesis and healing for patients. Surgeons have also benefitted—better cameras, tools with dexterity surpassing that of the human hand, and improved ergonomics have all been made possible with the robotic surgical system. However, the revolutionary nature of robotic surgery extends beyond the robot itself. This new platform has brought with it changes in surgical training and education, principally through the utilization of VR. Accurate depictions of human anatomy seen through augmented reality allow the surgeon-in-training to learn, practice and perfect their skills before they operate on their first patient. However, the anatomical knowledge required for a minimally invasive surgeon differs in many important ways from the knowledge gained through traditional prosection and dissection in the lab setting. These traditional methods of teaching human anatomy cater directly to a surgeon operating in an open setting and allow the student to learn anatomical structures using spatial relationships between different structures that are not easily translatable to a minimally invasive approach. Therefore, the anatomy curriculum must adapt to a world where less invasive surgical procedures are becoming the standard of care. It is integral that surgeons who operate in such an environment are equipped with accurate and relevant knowledge to ensure better outcomes for their patients.

It is this need for updated surgical training that surgical anatomists can best impart their skills and expertise. The anatomist should have a central role in modern anatomical education starting from the dissection labs of medical schools. Updating current training of surgical anatomists to incorporate the differences between how structures look in open and minimally invasive surgery would greatly benefit students who will enter the medical field at a time when minimally invasive surgery will be even more prevalent than it is now. In addition, the surgical anatomist has the ability to greatly influence the development and perfection of anatomy-based training for the robotic surgeon while working alongside the surgeon and software engineers trained in the use of VR. This collaboration and the creation of a new curriculum for robotic prosection and dissection can allow for extremely relevant and practical training that will develop surgical skills in a controlled environment. In an age where technology is revolutionizing the medical field, it is the surgeon and the anatomist who must rediscover human anatomy.

## Figures and Tables

**Figure 1 ijerph-18-12744-f001:**
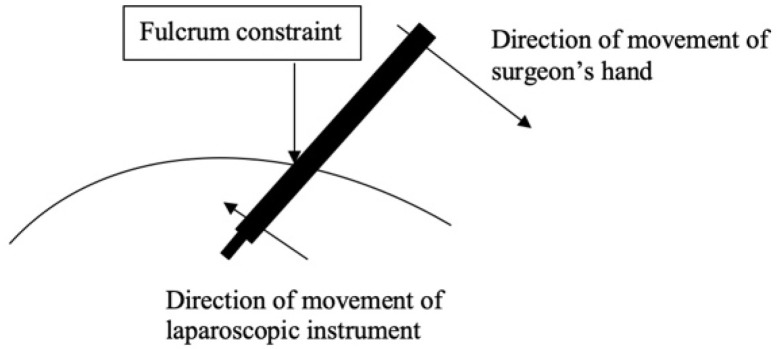
The fulcrum effect and its application to laparoscopic surgery. The surgical tool moves in the opposite direction of the surgeon’s hand due to the pivot point (fulcrum).

**Figure 2 ijerph-18-12744-f002:**
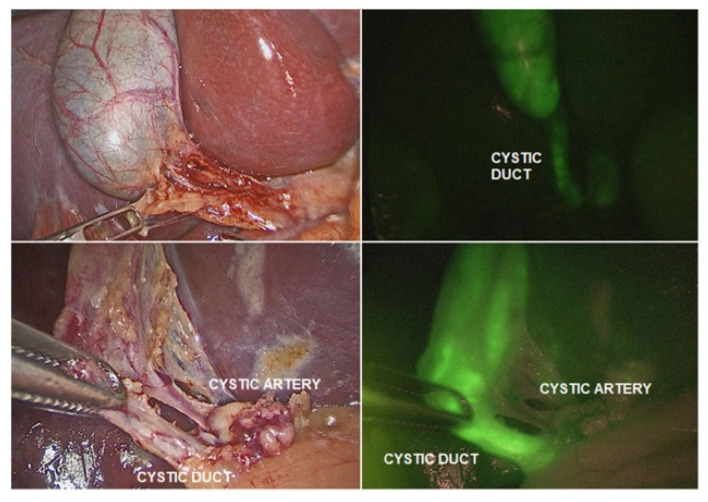
From Esposito et al. (2020) [10]. Indocyanine green (ICG)-guided near-infrared fluorescence (NIRF) identifying the cystic duct and artery.

**Figure 3 ijerph-18-12744-f003:**
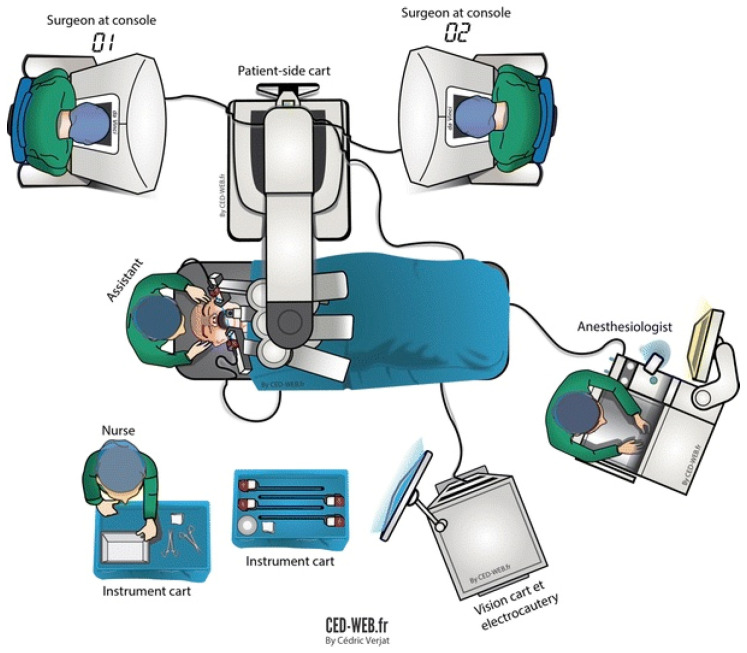
From Gorphe et al. (2017) [15]. Diagram of the operating room.

**Figure 4 ijerph-18-12744-f004:**
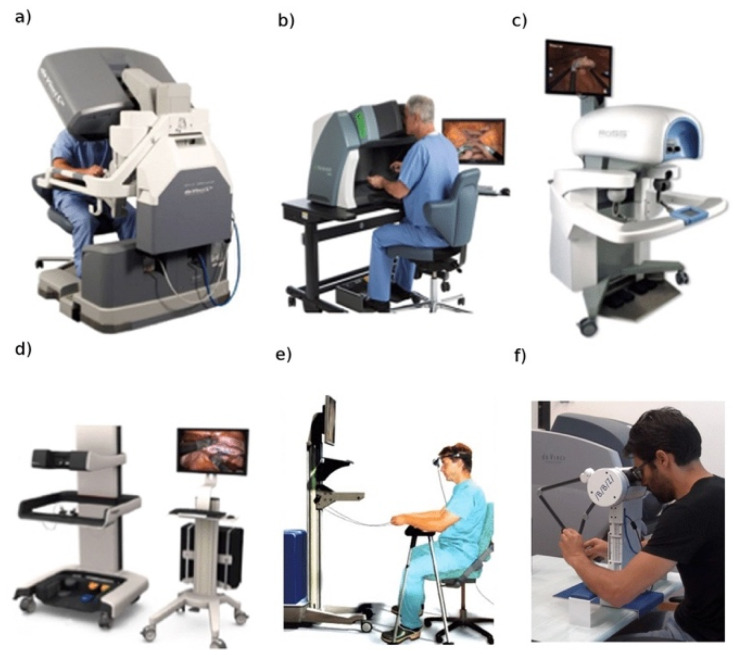
From Elek et al. (2019) [16]. Various virtual reality robotic surgery simulators. (**a**) Da Vinci Skills Simulator. (**b**) dV-Trainer. (**c**) Robotic Surgery Simulator. (**d**) Robotix Mentor. (**e**) SEP Robot. (**f**) Actaeon Robotic Surgery Training Console.

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
