# Peer review of "Robotic Surgery: Rediscovering Human Anatomy"

_ijerph, 2021, doi:10.3390/ijerph182312744_

Round 1

Reviewer 1 Report

While I appreciate this manuscript is listed as a commentary piece, the writing style is very casual/verbal throughout.

The only content which is covered in depth/detail is the Da Vinci operating system. The virtual reality section only contains 3 references, 2 of which are for examples of equipment. Considering the recommendations in the conclusion related to medical anatomical education there is a worrying lack of literature related to VR included in this review. Unfortunately the lack of references, despite a wealth of references being available, means that the conclusions cannot be justified.

The conclusion includes strong recommendations 

Author Response

Thank you for taking the time to review our paper and to provide insightful comments that have greatly helped us improve the quality of our work and support our arguments. The following are the changes we have made to the previous version of the paper under the guidance of your suggestions. We hope they are satisfactory.

Reviewer 1

“While I appreciate this manuscript is listed as a commentary piece, the writing style is very casual/verbal throughout.”

  • Thank you so much for this comment and for highlighting the delicate balance between casual language and a professional piece submitted to a journal. After going through the paper, the most colloquial language has been edited out so that the piece sounds less verbal. Most of these edits were to make the paper sound less conversational and more analytic. Examples have been included:
  • “Their manipulation is constrained by the so called “fulcrum effect” as the tip of the instrument moves in the opposite direction of the surgeon's hand that is holding the laparoscopic instrument” to Their manipulation is constrained by the “fulcrum effect” as the tip of the instrument moves in the opposite direction of the surgeon's hand that is holding the laparoscopic instrument (line 39)
    • We hope that this part of the paper, even with a small exclusion, now sounds much more direct and objective.
  • “When these students begin their training in robotic surgery, their entire perspective changes. Literally. Now localization of the target organ requires…” was changed to When these students begin their training in robotic surgery, their entire perspective changes. Now localization of the target organ requires…
    • The deletion of “literally” is intended to make this part of the paper much less casual than previously.
  • There are many other examples of these changes in the paper that we hope have succeeded in creating a more serious tone to our piece. Thank you again for your suggestion.

“The only content which is covered in depth/detail is the Da Vinci operating system. The virtual reality section only contains 3 references, 2 of which are for examples of equipment. Considering the recommendations in the conclusion related to medical anatomical education there is a worrying lack of literature related to VR included in this review. Unfortunately the lack of references, despite a wealth of references being available, means that the conclusions cannot be justified.”

  • Thank you so much for these comments. We absolutely admit that the VR section of the paper is lacking in references to support the stated claims. However, as you stated, evidence does exist that validates the use of surgical simulation in education. We have included more of these references in lines 164-169. These references validate the use of surgical simulation in the context of surgical training, physician skill development, and translation of skills to the OR setting. Our intention in including them is to justify the utility of virtual reality as a good and effective educational tool that can prepare surgeons for cases without posing a risk to a potential patient.
  • Citations regarding VR’s utility in surgical planning (line 176) and use in gaining surgical anatomy knowledge (line 179) have been added as well. These inclusions were added as a more concrete example of anatomical knowledge, required during surgical planning and active recall of anatomy knowledge, can be acquired through the use of virtual reality software.
  • We hope you can see an improvement in the scientific backing for the conclusions presented in our paper. After seeing this version of the virtual reality section, we are also much happier with the amount of data that we could find to support VR as a feasible and effective way to teach students and physicians. Thank you again, this comment particularly has improved the quality of our work greatly.

Reviewer 2 Report

Robotic Surgery: Rediscovering Human Anatomy

Would include a citation or two regarding the “long and steep” learning curve of laparoscopy (page 2, line 45)

Would include company headquarters (Sunnyvale, CA) (page 2, line 54) for Intuitive Surgical.

Could exchange citation for company website to pubmed citation – Dobbs et al “Single-port robotic surgery: the next generation of minimally invasive urology” (page 2, line 58)

Burgeoning not bourgenoning (page 2, line 69)

Loupes not loups (page 2, line 76)

Could include some historical context that one of the initial purposes of robotics was to facilitate surgical intervention from a distance for the US Military. (page 2, line 86)

Article is well written and compelling – some of the language veers towards informational materials but agree with the overarching viewpoint of robotics with the perspective of changing anatomy training in the future.

Author Response

Thank you for taking the time to review our paper. We are greatly appreciative of the time and work that it required to analyze and comment on the paper as well. We have taken the suggestions you provided very seriously and have addressed them point by point below. We hope they are satisfactory.

Reviewer 2

“Would include a citation or two regarding the “long and steep” learning curve of laparoscopy (page 2, line 45)”

  • Thank you for this comment ­– it is definitely a necessary addition. In this new version of the paper, we included two citations, one about the learning curve of laparoscopic cholecystectomy and the other about laparoscopic hernia repair. Both detail a higher rate of complications in the initial stages of training that resolve with more experience. (line 57)

“Would include company headquarters (Sunnyvale, CA) (page 2, line 54) for Intuitive Surgical.”

  • This change is reflected in the updated document (line 65)

“Could exchange citation for company website to pubmed citation – Dobbs et al “Single-port robotic surgery: the next generation of minimally invasive urology” (page 2, line 58)”

  • Thank you for the extremely helpful citation – we’ve changed the previous citation to the Dobbs et al paper (line 70).

“Burgeoning not bourgenoning (page 2, line 69)”

  • This change is reflected in the updated document (line 98)

“Loupes not loups (page 2, line 76)”

  • This change is reflected in the updated document (line 103)

“Could include some historical context that one of the initial purposes of robotics was to facilitate surgical intervention from a distance for the US Military. (page 2, line 86)”

  • Reference to the origins of the robotic approach by the Department of Defense have been added to the paper (lines 113-115). This is a very helpful piece of context that fits very well with the part of the paper you suggested. Thank you very much for your comment.

“Article is well written and compelling – some of the language veers towards informational materials but agree with the overarching viewpoint of robotics with the perspective of changing anatomy training in the future.”

  • Thank you so much for your kind words.

Reviewer 3 Report

The paper presents a discussion on the relationship between the advent of robotics and VR in surgery and the knowledge on anatomy as is currently acquired by surgeons during their academic training. The authors argue that a new way of teaching anatomy has to be developed, to improve the education of surgeons operating with robotic tools.

I believe that the discussion is interesting and valuable for publication. As a minor concern, it seems to me that the Zimmer Biomet's Rosa mentioned in the introduction is not directly comparable to the Intuitive Da Vinci, because the former is not designed for laparoscopic operation as it is instead the latter. I would suggest to mention the Medtronic Hugo as a reasonable competitor of the Da Vinci.

Author Response

Thank you for taking the time to read and review our paper. We are also very appreciative of the comments you provided, which were very thoughtful, and we hope the edits enclosed are satisfactory.

Reviewer 3

The paper presents a discussion on the relationship between the advent of robotics and VR in surgery and the knowledge on anatomy as is currently acquired by surgeons during their academic training. The authors argue that a new way of teaching anatomy has to be developed, to improve the education of surgeons operating with robotic tools.

I believe that the discussion is interesting and valuable for publication.

  • Thank you so much for your positive and encouraging comments.

As a minor concern, it seems to me that the Zimmer Biomet's Rosa mentioned in the introduction is not directly comparable to the Intuitive Da Vinci, because the former is not designed for laparoscopic operation as it is instead the latter. I would suggest to mention the Medtronic Hugo as a reasonable competitor of the Da Vinci.

  • Your advice to use Medtronic’s Hugo system instead of Zimmer Biomet’s was extremely helpful and we have made this change in the updated version of this paper (lines 62-63). We agree that it is better fit for the context of the conclusions we are presenting and thank you for the insightful suggestion.

Round 2

Reviewer 1 Report

The manuscript is now much improved. The language and depth of content is now appropriate for a commentary.